# Origin Intelligent Identification of *Angelica sinensis* Using Machine Vision and Deep Learning

**Zimei Zhang** [1], **Jianwei Xiao** [2], **Shanyu Wang** [1], **Min Wu** [1], **Wenjie Wang** [3], **Ziliang Liu** [1,*] and **Zhian Zheng** [1,*]

[1] College of Engineering, China Agricultural University, Beijing 100083, China
[2] Beijing Institute of Aerospace Testing Technology, Beijing 100074, China
[3] Chinese Academy of Agricultural Mechanization Sciences, Beijing 100083, China
[*] Correspondence: lzlcaucoe@163.com (Z.L.); zhengza@cau.edu.cn (Z.Z.)

**Abstract:** The accurate identification of the origin of Chinese medicinal materials is crucial for the orderly management of the market and clinical drug usage. In this study, a deep learning-based algorithm combined with machine vision was developed to automatically identify the origin of *Angelica sinensis* (*A. sinensis*) from eight areas including 1859 samples. The effects of different datasets, learning rates, solver algorithms, training epochs and batch sizes on the performance of the deep learning model were evaluated. The optimized hyperparameters of the model were the dataset 4, learning rate of 0.001, solver algorithm of rmsprop, training epochs of 6, and batch sizes of 20, which showed the highest accuracy in the training process. Compared to support vector machine (SVM), K-nearest neighbors (KNN) and decision tree, the deep learning-based algorithm could significantly improve the prediction performance and show better robustness and generalization performance. The deep learning-based model achieved the highest accuracy, precision, recall rate and F1_Score values, which were 99.55%, 99.41%, 99.49% and 99.44%, respectively. These results showed that deep learning combined with machine vision can effectively identify the origin of *A. sinensis*.

**Keywords:** *Angelica sinensis*; origin identification; deep learning; machine vision





## 1. Introduction

Origin identification plays a crucial role in establishing an orderly and controllable market for traditional Chinese medicine (TCM) and serves as a fundamental aspect of modernizing and advancing the TCM industry [1]. It is mainly used to effectively distinguish between good and bad medicine. Some authentic and high-quality Chinese medicines grown in specific areas are known as *Daodi* medicinal materials (DMMs), which have been proven to show conspicuous medical effects [2]. DMMs are the most effective materials [3,4] in CMM and are considered the essence of Chinese cultural heritage [5]. Nevertheless, the market for Chinese medicinal materials is still quite chaotic. Some producers usually mix Chinese medicinal materials from different origins so as to obtain more benefits. The recognition of DMMs is usually based on surface features. This approach relies on experience and assumptions to some extent [6], which are difficult for consumers to identify accurately. Moreover, the Chinese Society of Traditional Chinese Medicine does not provide useful guidance to help people to identify the origins of TCM. As a result, the market for Chinese medicinal materials and their origins are more likely to be mixed up. This harms the reputation of authentic medicinal materials, market supervision, effective procurement and the health of ordinary people, and hinders effective quality control of TCM. Therefore, it is of great significance for the quality control of traditional Chinese medicine to distinguish DMMs from medicinal materials of other origins effectively.

*Angelica sinensis* (*A. sinensis*) was first identified in the Shennong Materia Medica during the late Eastern Han Dynasty [7]. Its root is considered one of the noteworthy species in the genus Angelica [8,9]. *A. sinensis* contains a rich composition of polysaccharides, ferulates, essential oils, flavonoids, organic acids and coumarins, which was widely

consumed as a medicinal herb in both food and medicine [10]. In addition, *A. sinensis* is applied to treat blood deficiency, menstrual disorders and constipation clinically [11,12]. Since ancient times, *A. sinensis* from Minxian County has been considered to be a DMM. However, the current market for *A. sinensis* remains disordered. The complex and irregular appearance of *A. sinensis* makes it more difficult to conduct an accurate identification of its origin. Consequently, there is an urgent need to develop fast and effective methods for identifying the origin of *A. sinensis*.

Currently, the main methods used to identify the origin of *A. sinensis* include fingerprint identification [13–17], gene identification [6,18,19] and electronic nose bionic identification [20–22]. Though fingerprint technology is effective in authenticating the iconic component's authenticity, it fails to discern quality differences among regions [17]. The method of fingerprint identification is relatively simple and can encounter difficulties when evaluating the overall performance, resulting in the weak applicability of this method [23]. Although high-performance liquid chromatography (HPLC) can accurately measure the content of bioactive components of *A. sinensis*, the detection process is expensive and time-consuming [24]. In addition, near-infrared spectroscopy and electronic nose technology can rapidly detect the quality of *A. sinenis* in a non-destructive manner. However, near-infrared spectroscopy or the electronic nose method cannot distinguish the origin of *A. sinensis* due to the complex outline structure. Additionally, the associated equipment tends to be expensive [24].

With the advancement of modern science and technology, some researchers have focused on the objective quantification and evaluation of *A. sinensis'* appearance in order to inherit the essence of traditional methods and make up for the deficiency of human experience. For example, Wang and Yang identified *A. sinensis* from Heqing in Yunnan, Minxian County in Gansu, Bagchang in Gansu and Zhangxian County in Gansu through character and microscopic identification. They found that the surface of *A. sinensis* from Minxian County was dark yellowish brown, the cross section was yellowish white, the dotted secretion cavity was numerous and the roots were disordered, scattered and rich in aromas. The quality of *A. sinensis* from Minxian County was notably superior to that from other regions [25]. Wang et al. obtained the structure–texture images of *A. sinensis* from three different origins, including Minxian and Qinghai counties in Gansu Province and Heqing in Yunnan Province [24]. They established a support vector machine (SVM)-based prediction model to distinguish the origin of *A. sinensis*, and its prediction accuracy reached 98.49%. Despite all this, the image information collected from the three origins were still limited, since there are more than eight sources of origin in China. Therefore, further efforts are required to enhance the robustness and generalization ability of the developed SVM model.

In recent years, deep learning has been shown to automatically extract the original features of the data without any human interference, greatly improving the accuracy of the model [26]. It has become the most widely used computational method in the field of machine vision. Deep learning combined with machine vision showed good adaptability in different working scenes [27]. Convolutional neural networks (CNNs) are some of the most popular and used deep learning networks [28]. Xception is a lightweight CNN network with few parameters and good performance [29]. Therefore, this study aims to develop a robust deep learning method to identify the origin of *A. sinensis* by combining images with the Xception lightweight deep learning network, and establish a scientific and simple quality assurance approach for *A. sinensis* market sales. The commonly used methods, including support vector machine (SVM), K-nearest neighbors (KNN) and decision tree, were considered in a comparison to evaluate the performance of the developed model. The deep learning-based algorithm combined with a machine system is expected to be implemented directly on mobile phones to identify the origin of *A. sinensis* with the advantages of low power consumption, flexible use and convenient portability, which can better meet the needs of rapid detection in the field.

## 2. Materials and Methods

### 2.1. Samples of Angelica sinensis

A total of 1859 samples of *A. sinensis* were obtained for this study, sourced from various counties in different provinces of China. The samples were purchased from Gansu province, Min county (MX); Gansu province, Lintan county (LT); Gansu province, Wudu county (WD); Hubei province, Enshi county (HB); Yunnan province, Zhanyi county (YN); Sichuan province, Guangyuan county (SC); Qinghai province, Huangyuan county (QH); and Chongqing province, Chengkou county (CQ) (Figure 1). The characteristics of the *A. sinensis* samples are summarized in Table 1.

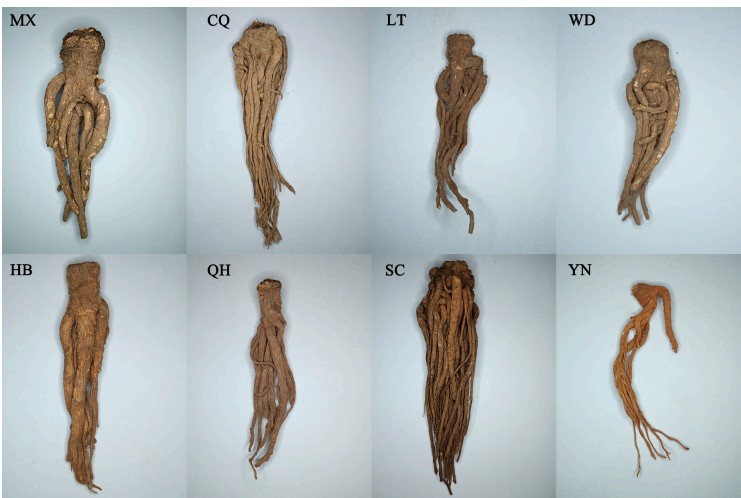

**Figure 1.** Images of *A. sinensis* from various origins.

**Table 1.** Characteristics of *A. sinensis* samples.

| Origin | Collection Time | The Number of Samples/pcs | The Total Number of Samples/pcs |
|--------|-----------------|---------------------------|----------------------------------|
| MX | 2021 | 397 | |
| LT | 2021 | 235 | |
| WD | 2021 | 197 | |
| HB | 2021 | 280 | |
| YN | 2021 | 231 | 1859 |
| SC | 2021 | 61 | |
| QH | 2021 | 158 | |
| CQ | 2021 | 300 | |

### 2.2. Image Acquisition

To ensure the acquisition of high-quality images of dried *A. sinensis*, a meticulously designed computer vision system was employed, as depicted in Figure 2. This advanced system comprised several key components, including a CCD industrial camera sourced from Basler, Germany (model number: aca250014-gc), an FA lens from Japan (ComputarM1214-MP2) and an LED light from Shanghai Jiaken Optoelectronics (JKVR-170W). The Basler camera utilized in this study is an exceptional 2.2-megapixel RGB camera, boasting a remarkable resolution of 2590 × 1942. Equipped with a cutting-edge CMOS sensor, this camera offers a maximum frame rate of 14 frames per second (fps). Moreover, its effective operating temperature range spans from 0 to 50 °C, making it suitable for various environmental conditions. To minimize any potential interference caused by external reflections, the outer frame of the entire image acquisition system was diligently enveloped with non-reflective black fabric. Conversely, the side facing the *A. sinensis* samples remained open, allowing for precise placement and ideal imaging conditions. For the purpose of capturing and storing the images of the samples, OpenCV software (version 3.0) was seamlessly

integrated into the system. Leveraging the capabilities of OpenCV, this computer vision system efficiently captured and saved the acquired images, ensuring their accessibility and usability for subsequent analysis.

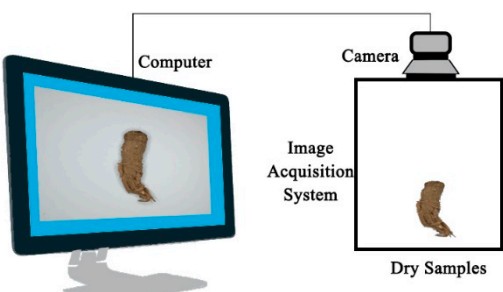

**Figure 2.** Image acquisition system.

During the meticulous image acquisition process, samples sourced from various origins were positioned precisely at a fixed distance of 40 cm from the camera. To ensure a consistent and uniform background, a sheet of white paper was carefully placed on the loading platform. Subsequently, the samples were systematically arranged horizontally upon this pristine white surface. To guarantee precise measurement and facilitate subsequent analysis, a ruler adorned with a detailed scale was meticulously positioned to the right of each *A. sinensis* sample, maintaining a specified distance for accurate reference. In order to achieve optimal lighting conditions for capturing clear and informative images, the LED lamp brightness was calibrated with utmost care to a specific value of 1210. This ensured that the desired level of brightness was achieved while minimizing any potential adverse effects such as overexposure or shadows. The resultant images were captured with remarkable resolution, boasting an impressive size of 3120 pixels by 4160 pixels. To capture all notable features, photographs of both the front and back sides of each sample were diligently taken. The outcome of this rigorous process yielded a grand total of 3718 meticulously crafted images. Allocating these images appropriately, 70% of the dataset was designated for training purposes, while reserving 15% for verification and another 15% for testing.

*2.3. Preprocessing*

Preprocessing plays a crucial and indispensable role in significantly enhancing the quality of images within a computer vision system. Among the various components of preprocessing, data augmentation stands out as a pivotal step in training neural networks to effectively address the challenge of overfitting [30]. The data enhancement process can be categorized into two forms: offline data augmentation and online dynamic data augmentation.

Data enhancement in the offline phase commonly employs a variety of techniques, with random transformations such as rotation, translation and flipping being the most prevalent. Additionally, modifications to the hue, brightness and saturation of the data are frequently employed. These methods greatly contribute to increasing the diversity of the dataset, thereby expanding its representation and enriching the training process. By generating additional training samples through these operations, the neural network is fortified and better equipped to handle various scenarios.

Dynamic data enhancement, on the other hand, is typically implemented through the utilization of software tool functions within deep learning frameworks. During the training phase, images undergo random conversions on a per-cycle basis. This dynamic approach ensures that the data are continuously enhanced, effectively combating the risk of overfitting. By subjecting the data to random conversions in each training cycle, the network is exposed to a broader range of variations, preventing it from memorizing specific instances and facilitating its ability to generalize to new and unseen data.

In this experiment, a comprehensive approach utilizing both offline and online dynamic data enhancement was used to amplify the dataset and prevent over-fitting. The

offline data enhancement methods and enhancement effect diagram are shown in Figure 3. Online enhancement is the random scaling (0.1–1.1 times) of the data during training to enhance the generalization of the model. By combining both offline data enhancement techniques and online dynamic data enhancement strategies, the preprocessing stage optimizes the quality of the dataset and empowers the neural network to learn robust and generalizable representations from the available data. This comprehensive approach significantly contributes to mitigating overfitting and enhancing the overall performance of computer vision systems.

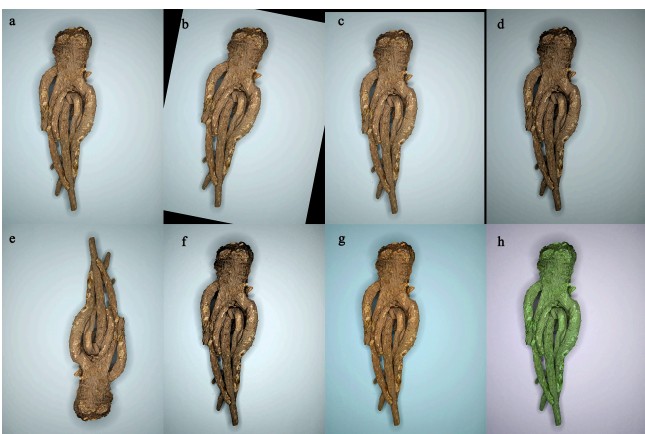

**Figure 3.** Example images of data augmentation: (**a**) original image, (**b**) rotated image, (**c**) translated image, (**d**) brightness-transformed image, (**e**) flip image, (**f**) contrast-transformed image, (**g**) saturation-transformed image, and (**h**) chroma-transformed image.

### 2.4. Origin Recognition of A. sinensis

Convolutional neural networks (CNNs) were first introduced by Fukushima [31]. CNN has two significant advantages: first, it does not need to preprocess the input layer data, which can reduce the computational workload of the experiment; second, in the training process, CNN automatically extracts features to complete image classification [32]. CNN mainly simulates the human brain with the abilities of self-learning and self-regulation through the interconnection between neuron nodes. Each neuron has a learnable weight and bias. CNN consists of an input layer, an output layer and multiple hidden layers, among which the hidden layer is composed of the convolution layer, the pooling layer, the full connection layer (FC) and various normalized layers.

#### 2.4.1. Convolution Layer

The convolutional layer uses convolution operations to combine two sets of information and simulate the response of a single neuron to a visual stimulus. As the core of CNN, it undertakes most of the calculation work, extracts features, reduces the number of parameters to be trained and reduces the complexity of the deep network.

#### 2.4.2. Pooling Layer

The pooling operation is applied after convolution. It is used to reduce dimensions by correlating the output of a layer of neuronal clusters into a single neuron. The most commonly used methods in pooling operations are average pooling and maximum pooling.

#### 2.4.3. Fully Connected Layer

The fully connected layer resides at the end of the CNN architecture. Its main function is to further extract high-level features from the features extracted from the convolutional layer, so as to achieve the purpose of classification. Softmax logistic regression is used to classify and recognize images.

2.4.4. Xception Architecture

Xception [33] is an improved model of Inception-v3 proposed by Google, which uses deep separable convolution to replace the original convolution operation in Inception-v3, greatly improving the accuracy of the model. The essence of depth-separable convolution is to divide standard convolution into a deep convolution and a point-by-point convolution to reduce the computational complexity and the number of parameters of the model. The network contains a total of 36 convolutional layers, which are divided into 14 modules for feature extraction. The 14 modules are distributed between entry flow, middle flow and exit flow, where "Conv" stands for standard convolution, and "SepConv" stands for deeply separable convolution. The default picture input size for the Xception model is 299 × 299 and the default number of channels is 3.

For the consideration of hardware conditions and the classification of *A. sinensis* origin, the Xception model was improved by migration learning. Model fine-tuning is the process of unfreezing the top layers of the pre-training model so that the trained features more closely match the task at hand [34].

In the experiment of this paper, the fully connected layer was fine-tuned, and the original fully connected layer was directly removed and replaced with a new fully connected layer with output 8, so as to conform to the identification of 8 origins. The main structure of the fine-tuned Xception is shown in Figure 4. Augmented images were taken as input through the entry flow section, and then subsequently channeled into the middle flow section, where the feature map process was repeated 8 times; finally, they were channeled through the exit window.

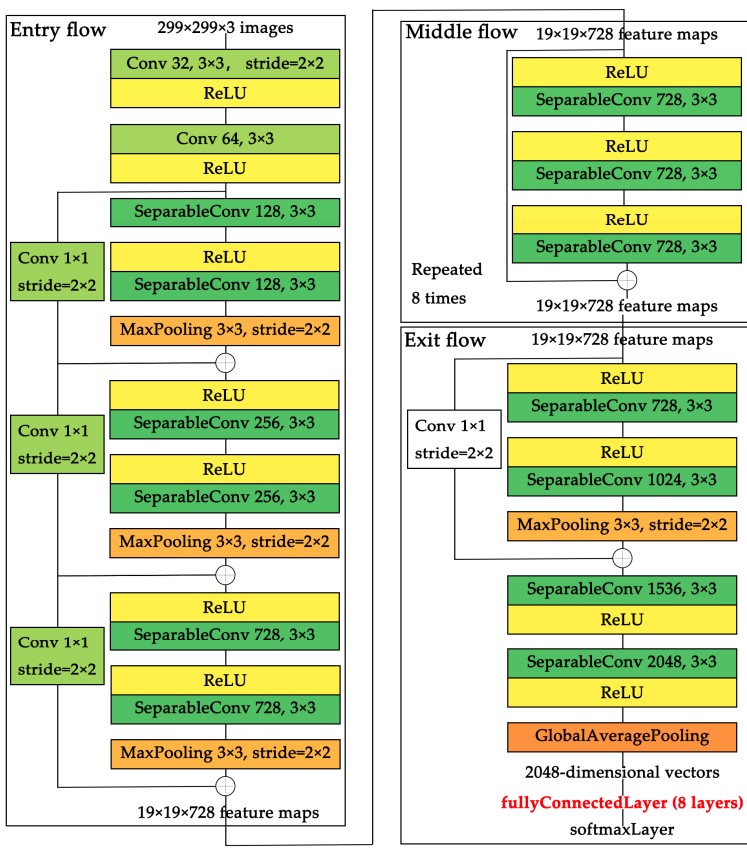

**Figure 4.** The body structure diagram of fine-tuned Xception.

*2.5. Identification Performance Evaluation*

The performance of the model in origin recognition of *A. sinensis* was evaluated by calculating the accuracy, recall, precision and F1_Score [26,35,36], which are shown in Equations (1)–(4).

$$Accuracy = \frac{TP + TN}{TP + FN + TN + FP} \tag{1}$$

$$Recall = TPR = \frac{TP}{TP + FN} \tag{2}$$

$$Precision = \frac{TP}{TP + FP} \tag{3}$$

$$F1\_Score = 2 \times \frac{Recall \times Precision}{Recall + Precision} \tag{4}$$

TP stands for true positive, which means that the data that are supposed to be true are actually true; FP stands for false positive, i.e., data that are predicted to be true are actually false; FN stands for false negative, representing data that are predicted to be false but are actually true; and TN stands for true negative, meaning data that are supposed to be false and actually are false.

## 3. Results and Discussion

The purpose of this study is to identify the origin of *A. sinensis* by combining images with the Xception lightweight deep learning network. Considering the optimal model construction, the effects of the number of datasets, learning rate, solver, epoch and batch size on the accuracy were investigated, and the optimal parameters were determined. In order to evaluate the developed CNN model, the results of the best model test results were compared with traditional classification models such as SVM, KNN and decision tree. The features used in the traditional classification models of SVM, KNN and decision tree are color features, texture features and shape features. All algorithms were performed on MATLAB(R2021b) software.

*3.1. Model Building*

3.1.1. Determination of the Dataset

The size of the dataset plays a crucial role in determining the accuracy of a model. Inadequate training data can result in insufficient learning, while an excessive amount of data can lead to overfitting, both of which significantly impact the accuracy of model testing. In this study, five different datasets with varying numbers of images were constructed by applying random rotations, flips, translations and adjustments to chromaticity, contrast and brightness of the data (as summarized in Table 2). These diverse datasets were then utilized to train the model, and the accuracy of the validation set was evaluated, as depicted in Figure 5.

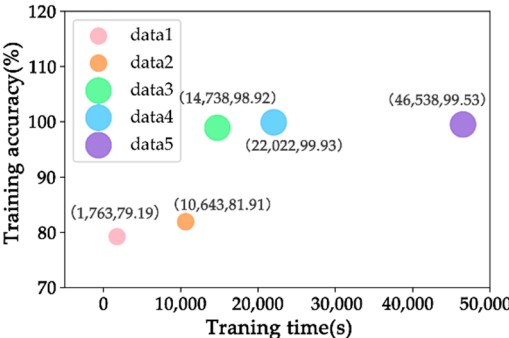

**Figure 5.** Influence of datasets on the training accuracy of CNN model.

**Table 2.** Composition of different datasets.

| Data | Image Transformation | Number of Training Sets/pcs | Number of Test Sets/pcs | Total Datasets/pcs |
|---|---|---|---|---|
| data1 | No | 2980 | 738 | 3718 |
| data2 | Rotate and flip randomly | 8940 | 2214 | 11,154 |
| data3 | Rotate, flip, and shift randomly | 17,880 | 4428 | 22,308 |
| data4 | Rotate, flip, translate and adjust chroma randomly | 26,820 | 6642 | 33,462 |
| data5 | Randomly rotate, flip, translate and adjust the chrominance, contrast, brightness | 80,460 | 19,926 | 100,386 |

Figure 5 clearly demonstrates that the accuracy varies significantly across the different datasets. As the dataset size increases, so does the accuracy of the model. However, it is important to note that larger datasets also require longer computation time. Hence, in order to strike a balance between accuracy and computational efficiency, the optimal sample size for this study was determined to be 33,462. By carefully selecting and optimizing the dataset size, the model achieved a desirable level of accuracy without compromising on computational efficiency. This finding has practical implications for designing efficient and effective machine learning systems in various domains.

### 3.1.2. Learning Rate Determination

Through extensive literature references [37,38], the initial learning rate of the training model was investigated at 0.0001, 0.001, 0.01, 0.05 and 0.1, respectively, and its verification set accuracy is shown in Figure 6. When the learning rate was 0.0001, the accuracy was the highest, i.e., 99.6%. Therefore, 0.0001 was selected as the appropriate learning rate.

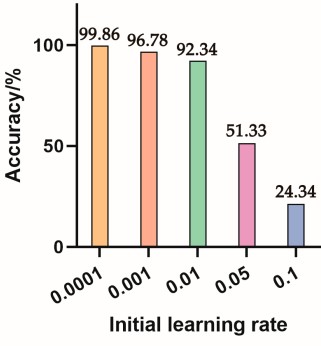

**Figure 6.** Influence of initial learning rate on the training accuracy of CNN model.

### 3.1.3. Determination of Optimization Method (Solver)

The training model was based on the sdgm, adam and rmsprop algorithms. Sdgm [39] is a stochastic gradient descent algorithm that utilizes momentum to reduce oscillation, expedite convergence and escape local optima. Furthermore, rmsprop [40] and adam [41] are adaptive learning rate optimizers [39] that dynamically adjust the learning rate for each parameter. This adaptive adjustment enables gradual reduction of update steps for different parameters over time, leading to more precise parameter adjustments. The accuracy of its verification set is shown in Figure 7.

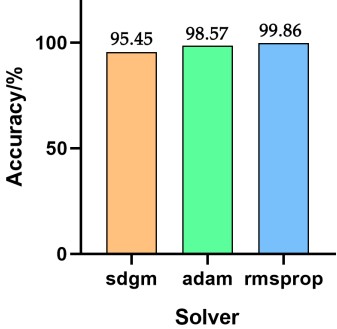

**Figure 7.** Influence of optimization method on the training accuracy of CNN model.

It can be seen from Figure 7 that the rmsprop algorithm obtained the highest training accuracy of 99.86%. Therefore, the rmsprop algorithm was selected as the optimization algorithm for subsequent training.

### 3.1.4. Determination of Training Epochs

The training epochs represent the number of epochs needed for data. As shown in Figure 8, the accuracy was linearly increased with the increase of training epochs from one to six. When the number of training epochs was set at six, the highest accuracy rate was 99.89%. Therefore, the use of six training epochs was selected for the training model.

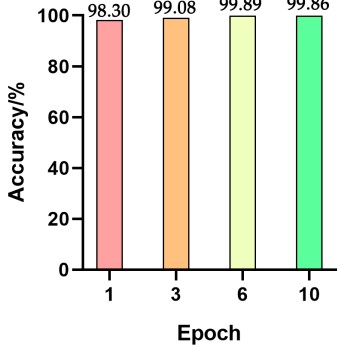

**Figure 8.** Influence of training epochs on the training accuracy of CNN model.

### 3.1.5. Batch Size Determination

In the training process of a convolutional neural network, the batch size will affect the memory configuration and accuracy of the computer. As shown in Figure 9, the highest accuracy was seen with the batch size of 20, i.e., 99.93%. Therefore, a batch size of 20 was selected for the training process.

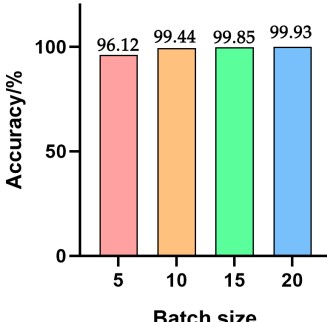

**Figure 9.** Effect of different batch sizes on the training accuracy of CNN model.

### 3.2. Model Training

After repeated training of the model, the model parameters of this training set were as follows: the dataset was data4, the initial learning rate was 0.001, the optimization method was rmsprop, the number of training epochs was 6 and the batch size was 20. Under such conditions, the accuracy and loss changes of the training set and verification set in the training process of the training model are shown in Figure 10.

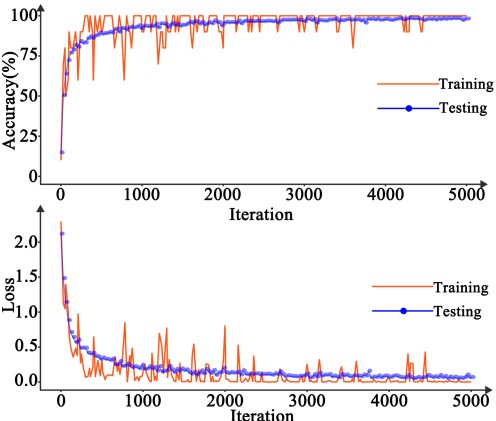

**Figure 10.** Accuracy and loss changes of training set and verification set in origin identification.

As can be seen from Figure 10, the accuracy curve and loss curve of fine-tuned Xception were better when the above optimization parameters were adopted. When the number of iterations reached 3000, the prediction accuracy was close to 100%. When the number of iterations reached 4000, the changes of accuracy value tended to be gentle and stable. The change trend of loss values was opposite to that of accuracy values.

### 3.3. Model Testing

In order to evaluate the predictive performance of the model, it is necessary to analyze the evaluation indicators of the test set of samples. During the whole network learning process, the test datasets used the performance of the model. The test results of a single image are shown in Figure 11, and the corresponding confusion matrix of origin identification results is shown in Figure 12.

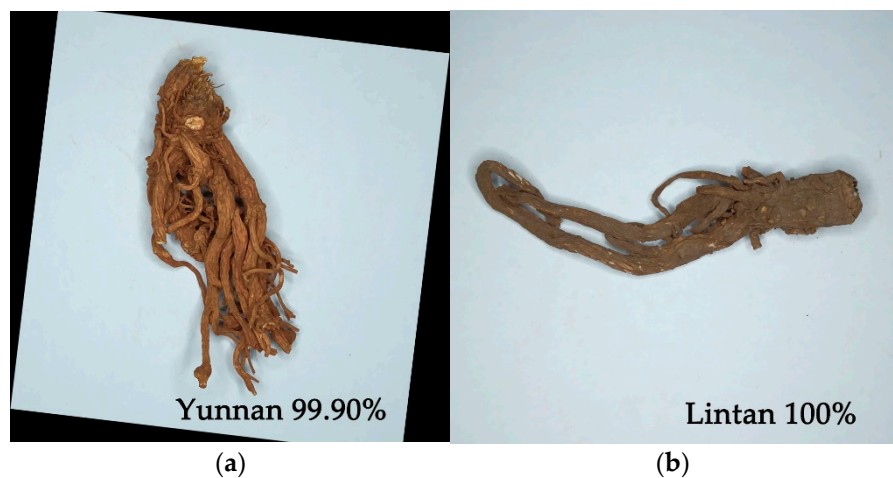

(**a**)　　　　　　　　　　　　　　　(**b**)

**Figure 11.** Matching degree of single image origin identification. (**a**) Yunnan (99.90%). (**b**) Lintan (100%).

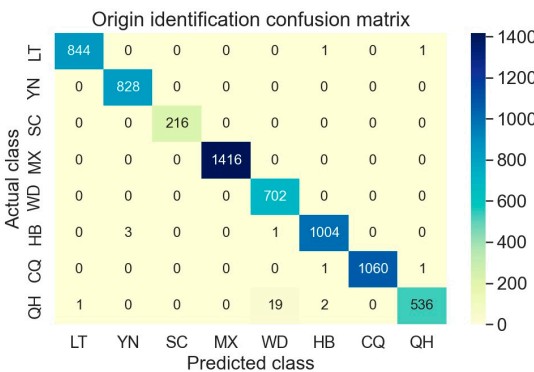

**Figure 12.** The confusion matrix of *A. sinensis* origin identification.

As shown in Figure 11, the accuracy of the fine-tuned Xception model in identifying the origin of *A. sinensis* based on the two random images was close to 100%. From Figure 12, it can be seen the *A. sinensis* from YN, SC, MX and WD were all correctly identified. Only 0.2% of LT were identified as HB and QH. Only 0.4% of HB were misidentified as YN and WD. Only 0.2% of CQ were wrongly predicted to be HB. The predicted error of model for the QH samples was the highest, i.e., 3.9%; most of them were wrongly predicted to be WD.

The accuracy, recall rate and F1_Score calculated by the confusion matrix are shown in Table 3.

**Table 3.** Results of origin identification test.

| Origin | Accuracy/% | Precision/% | Recall/% | F1_Score/% |
|---|---|---|---|---|
| Total accuracy | 99.55 | 99.41 | 99.49 | 99.44 |
| LT | 99.76 | 99.80 | 99.90 | 99.85 |
| YN | 100.00 | 100.00 | 99.60 | 99.80 |
| SC | 100.00 | 100.00 | 100.00 | 100.00 |
| MX | 100.00 | 100.00 | 100.00 | 100.00 |
| WD | 100.00 | 100.00 | 97.20 | 98.58 |
| HB | 99.60 | 99.60 | 99.60 | 99.60 |
| CQ | 99.81 | 99.80 | 100.00 | 99.90 |
| QH | 96.06 | 96.10 | 99.60 | 97.82 |

As can be seen from Table 3, the identification accuracy, precision, recall rate and F1_Score of each origin were all above 95%, which indicated that the developed model could effectively identify the local origin of *A. sinensis* with good performance.

In order to further evaluate the performance of the CNN model, the support vector machine (SVM), K-nearest neighbors (KNN) and decision tree methods were developed as a comparison. After conducting extensive optimization through 10-fold cross-validation for these three training models, their primary parameters were determined. In the case of SVM, a comparative evaluation of the linear, quadratic, cubic and Gaussian kernel functions revealed that the Gaussian kernel function yielded the most favorable outcomes. Consequently, it was chosen as the kernel function for SVM. Likewise, the multiple classification method was set to one-to-one, and the kernel scale was established at 25. Regarding KNN, performance comparisons were conducted among different numbers of nearest neighbors, such as 1, 10 and 100, resulting in the identification of 10 as the optimal number. Consequently, the number of nearest neighbors was set to 10, and the distance measure was determined based on the included angle cosine. Similarly, for the decision tree, the maximum split number was set to four following the same methodology.

The test results of CNN, SVM, KNN and decision tree are shown in Table 4. It can be seen from Table 4 that the recognition accuracy, precision, recall rate and F1_Score of the CNN model were 99.49%, 99.41%, 99.49% and 99.44%, respectively, which were higher than those of SVM, KNN and decision tree, while the accuracy, precision, recall rate and

F1_Score of the three methods of SVM, KNN and decision tree used in this study were all less than 50%. The observed phenomenon could be attributed to the intricate variations in the appearance of *A. sinensis* sourced from different origins, where discernible discrepancies in its visual characteristics among various production regions are limited. The challenge lies in the difficulty of manually extracting subtle distinguishing features, leading to the reduced fitting capabilities of SVM, KNN and decision tree algorithms. However, deep learning techniques excel in automatically extracting high-dimensional features through extensive data training, enabling effective identification of all regions.

**Table 4.** Comparison of test results of different models of origin identification.

| Models | Performance Metric | | | |
|---|---|---|---|---|
| | Accuracy/% | Precision/% | Recall/% | F1_Score/% |
| SVM | 35.88 | 29.61 | 33.93 | 33.02 |
| KNN | 35.71 | 37.36 | 34.33 | 31.86 |
| Decision tree | 28.9 | 19.69 | 16.76 | 18.11 |
| CNN | 99.55 | 99.41 | 99.49 | 99.44 |

In terms of identified performance, the results of the developed CNN method combined with machine vision were also better than those of identifying *A. sinensis* origin via fingerprint [15], electronic nose [22] and HPLC, as the prediction accuracies of these methods were all below 98%. On the other hand, the training model in this study only needs to be tested after completion once, and the identified results can be obtained rapidly based on the images of the materials. Compared to the fingerprint, electronic nose and HPLC methods, the established method in the current work can greatly reduce the identifying time and economic cost, which is expected to facilitate communication between producers and consumers [42].

## 4. Conclusions

In this study, a machine vision system was applied to capture images of *A. sinensis* from eight different origins. A fine-tuned Xception network, a CNN model, was proposed to identify the origins of *A. sinensis*. The experimental results revealed that the optimal hyperparameters for the dataset, initial learning rate, solver algorithms' training epoch and batch size were dataset 4, 0.001, rmsprop, 6, and 20, respectively, which obtained the best training performance. The augmented data were utilized to evaluate the CNN model. The identification accuracy, precision, recall rate and F1_Score for each origin surpassed 95%, indicating the effectiveness of the developed method in accurately identifying the local origin. Compared to the traditional classification models, the recognition accuracy, precision, recall rate and F1_Score of the CNN model were much higher than SVM, KNN and decision tree. The results indicate that the developed CNN model had better robustness and generalization performance. These results indicated that the combination of machine vision system with the Xception model is an effective and non-invasive method to distinguish the origin of *A. sinensis*, which may be useful for farmers and producers in determining the value of a product in different origins. More rhizome medicinal herbs will be considered to verify the performance of the model in the future, so as to increase application scopes of the origin identification of TCM.

**Author Contributions:** Conceptualization, Z.Z. (Zimei Zhang), Z.L. and Z.Z. (Zhian Zheng); methodology, Z.Z. (Zimei Zhang); resources, J.X. and Z.Z. (Zhian Zheng); experiment, Z.Z. (Zimei Zhang) and S.W.; programming, Z.Z. (Zimei Zhang) and W.W.; data curation, Z.Z. (Zimei Zhang); writing—original draft preparation, Z.Z. (Zimei Zhang); writing—review and editing, J.X., M.W. and Z.L.; funding acquisition, Z.Z. (Zhian Zheng). All authors have read and agreed to the published version of the manuscript.

**Funding:** This research was funded by the Innovation Team and Talents Cultivation Program of National Administration of Traditional Chinese Medicine (No: ZYYCXTD-D-202205), China Agriculture Research System (CARS-21) and China Postdoctoral Science Foundation (2022M723416).

**Institutional Review Board Statement:** Not applicable.

**Data Availability Statement:** Data are contained within the article.

**Conflicts of Interest:** The authors declare no conflict of interest.

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
