# Peer review of "Origin Intelligent Identification of Angelica sinensis Using Machine Vision and Deep Learning"

_agriculture, doi:10.3390/agriculture13091744_

Round 1
Reviewer 1 Report
The study describes the classification of Angelica sinensis plant used in traditional medicine of China by deep learning methods. The fact that the data set is new in the study is exciting for studies in this field. However, there are some academic shortcomings of the study. These deficiencies are listed as follows.
1- The image collection system given in Figure 2 should be given in accordance with its actual use if possible, or the image should be drawn again and the image on the computer and the image of the object in the system should be the same.
2- When the number of samples given in Table 1 is examined, why is the number of samples of type SC small while the numbers of external samples of type SC are close to each other?
3- Add a multi-class confusion matrix that you used in the study to the study by reviewing the following article for performance evaluation.
Source: https://www.hindawi.com/journals/cin/2022/2062944/
4- The features of your computer that you use in the study 3. specify it in the introductory paragraph of the section.
5- Why did you choose the Xception model when there are many CNN models? Have you tested other pre-trained networks? (VGG-16, VGG19 etc. like)
6- Give the performance metric names given in Table 4 in one line.
7- Why did you give the results as a percentage when there is no percentage expression (%) in the formulas you gave for performance metrics? Correct the formulas according to the results you have obtained.
8- considering the results given in the 24th reference, why did you get an accuracy value of 35% in your study, while the accuracy rate obtained by the SVM model is something like 98%?
9- Add the time spent for training and testing for the hyperparameters you changed in the study to the study. It should be easier for readers to Decipher the relationship between the time spent and classification accuracy according to the modified parameter.
Author Response
Dear Editor and the reviewers:
We sincerely thank you and the anonymous reviewers for your helpful comments, corrections, and suggestions on the previous version of this manuscript. We believe that our work has benefited substantially from the invaluable input of the review team. Below is the detail of how we have addressed the reviewer’s comments.
All the changes within manuscript are highlighted in red color, deleted words are not shown to avoid confusions. Moreover, the detailed corrections are listed below, point by point.
Refee#1
Comments and Suggestions for Authors
The study describes the classification of Angelica sinensis plant used in traditional medicine of China by deep learning methods. The fact that the data set is new in the study is exciting for studies in this field. However, there are some academic shortcomings of the study. These deficiencies are listed as follows.
1- The image collection system given in Figure 2 should be given in accordance with its actual use if possible, or the image should be drawn again and the image on the computer and the image of the object in the system should be the same.
Response:
Thanks very much for your suggestion. We revised it accordingly.
2- When the number of samples given in Table 1 is examined, why is the number of samples of type SC small while the numbers of external samples of type SC are close to each other?
Response:
Thanks very much for your suggestion. Due to the lower production of Angelica sinensis in Sichuan compared to other regions, the availability of samples is limited. Conversely, Minxian County, which is renowned as an authentic production area for Angelica sinensis, has the highest production volume, leading to a larger sample collection.
3- Add a multi-class confusion matrix that you used in the study to the study by reviewing the following article for performance evaluation.
Source: https://www.hindawi.com/journals/cin/2022/2062944/
Response:
Thank you for your valuable suggestion. We have cited the recommended literature as the 36th reference in our manuscript in line 205.
4- The features of your computer that you use in the study 3. specify it in the introductory paragraph of the section.
Response:
Thank you for your suggestion. We have incorporated the following sentence in lines 218-220: "The features used in the traditional classification models of SVM, KNN and decision tree are color features, texture features and shape features."
5- Why did you choose the Xception model when there are many CNN models? Have you tested other pre-trained networks? (VGG-16, VGG19 etc. like)
Response:
Thank you very much for your suggestion. We have provided an explanation for choosing Xception in lines 88-89: "Xception is a lightweight CNN network with few parameters and good performance".
6- Give the performance metric names given in Table 4 in one line.
Response:
Thanks very much for your suggestion. We revised it accordingly. The added citation is on line 205
7- Why did you give the results as a percentage when there is no percentage expression (%) in the formulas you gave for performance metrics? Correct the formulas according to the results you have obtained.
Response:
Thanks very much for your suggestion. We revised it accordingly.
8- considering the results given in the 24th reference, why did you get an accuracy value of 35% in your study, while the accuracy rate obtained by the SVM model is something like 98%?
Response:
Thank you very much for your valuable comment. In reference 24, SVM was employed to identify Angelica sinensis from four distinct regions, whereas our study focuses on identifying the plant from eight regions. In fact, the differences become more complex with the increase of the number of production areas, and thus lead to lower accuracy. Deep learning can automatically extract certain features that cannot be extracted by humans, thus improving accuracy. This is precisely why deep learning is crucial for improving accuracy. And our results have demonstrated the effectiveness of deep learning in identifying the origin of Angelica sinensis.
9- Add the time spent for training and testing for the hyperparameters you changed in the study to the study. It should be easier for readers to Decipher the relationship between the time spent and classification accuracy according to the modified parameter.
Response:
Thank you very much for your suggestion. Compared to the commonly used methods such as fingerprint identification, gene identification, and electronic nose bionic identification, the newly proposed deep learning-based identification method in this study has significant advantages. Notably, this method does not cause any damage to the samples and can provide real-time results within a relatively short timeframe. In regard to the origin identification of A. sinensis, we prioritizes model performance, followed by training time. Furthermore, incorporating the time dimension could introduce additional complexity in determining the optimal hyperparameters of the model.
Reviewer 2 Report
1. Paper focus on presenting a study using on identification of Angelica sinensis using machine vision and deep learning. The domain and topic of study is good.
2. Organization of the paper is good.
3. The study presented needs improvement in technical depth. Basics of proposed algorithm is clearly highlighted.
4. Novelty of proposed work needs improvement and clarity. The basic algorithm is also already existing as cited by the authors in Ref [13]. What is the research gap should be clearly highlighted.
5. Results and illustrations is well presented.
6. Good number of references is cited in the paper. Ref 33 has DOI which runs to next line. Authors need to focus on paper formatting. Authors may consider the paper below for citation in the journal.
V. K. G. Kalaiselvi, S. Hariharan, V. Kukreja, H. Venkateswarareddy, P. Hemanth and P. Dinesh, "Secured Cloud Environment for Improved Web Services in Agricultural Sector," 2023 7th International Conference on Intelligent Computing and Control Systems (ICICCS), Madurai, India, 2023, pp. 1406-1410, doi: 10.1109/ICICCS56967.2023.10142402.
7. Results comparison can be improved in the context of the proposed work.
Few places minor extensive corrections in grammar is required.
Author Response
Dear Editor and the reviewers:
We sincerely thank you and the anonymous reviewers for your helpful comments, corrections, and suggestions on the previous version of this manuscript. We believe that our work has benefited substantially from the invaluable input of the review team. Below is the detail of how we have addressed the reviewer’s comments.
All the changes within manuscript are highlighted in red colour, deleted words are not shown to avoid confusions. Moreover, the detailed corrections are listed below, point by point.
Refee#2
Comments and Suggestions for Authors
- Paper focus on presenting a study using on identification of Angelica sinensis using machine vision and deep learning. The domain and topic of study is good.
Response:
Thank you very much for your positive feedback and support.
- Organization of the paper is good.
Response:
Thank you very much for your positive feedback and support.
- The study presented needs improvement in technical depth. Basics of proposed algorithm is clearly highlighted.
Response:
Thank you for your valuable suggestion. We sincerely appreciate your advice.
Currently, there is a significant issue of misrepresentation and counterfeit Chinese medicinal materials, resulting in a chaotic situation regarding their origin. Common techniques such as fingerprint analysis, chemical composition testing, and genetic identification not only lead to losses in the medicinal materials but also fail to provide timely detection results. In contrast, machine vision combined with deep learning technology offers a non-destructive and efficient approach to obtain results. This technology has been widely applied in industrial and agricultural sectors. However, its application in solving the problem of origin identification in Chinese medicinal materials is still limited.
The algorithm utilized in our study, while relatively common, has demonstrated promising performance in identifying the origin of Angelica sinensis. In the future, we plan to further enhance the model's robustness by integrating intelligent optimization algorithms. Additionally, we intend to conduct extensive testing with a larger sample size to further improve the model's performance.
- Novelty of proposed work needs improvement and clarity. The basic algorithm is also already existing as cited by the authors in Ref [13]. What is the research gap should be clearly highlighted.
Response:
Thank you for your valuable suggestion. We sincerely appreciate your advice.
Currently, there is a significant issue of misrepresentation and counterfeit Chinese medicinal materials, resulting in a chaotic situation regarding their origin. Common techniques such as fingerprint analysis, chemical composition testing, and genetic identification not only lead to losses in the medicinal materials but also fail to provide timely detection results. In contrast, machine vision combined with deep learning technology offers a non-destructive and efficient approach to obtain results. This technology has been widely applied in industrial and agricultural sectors. However, its application in solving the problem of origin identification in Chinese medicinal materials is still limited.
Deep learning has demonstrated remarkable performance and has found extensive applications across various industrial sectors. However, when it comes to intricate Angelica materials, there is currently a lack of satisfactory methods for origin identification. This study provides compelling evidence that the integration of deep learning and machine vision can effectively discern the multiple origins of Angelica.
- Results and illustrations is well presented.
Response:
Thank you very much for your positive feedback and support.
- Good number of references is cited in the paper. Ref 33 has DOI which runs to next line. Authors need to focus on paper formatting. Authors may consider the paper below for citation in the journal. V. K. G. Kalaiselvi, S. Hariharan, V. Kukreja, H. Venkateswarareddy, P. Hemanth and P. Dinesh, "Secured Cloud Environment for Improved Web Services in Agricultural Sector," 2023 7th International Conference on Intelligent Computing and Control Systems (ICICCS), Madurai, India, 2023, pp. 1406-1410, doi: 10.1109/ICICCS56967.2023.10142402.
Response:
Thank you very much for your reminder. We have made the necessary changes according to the reference 33.
- Results comparison can be improved in the context of the proposed work.
Response:
Thank you very much for your reminder. We have addressed the issue of result comparison in lines 324-330 with the following explanation: "The observed phenomenon could be attributed to the intricate variations in the appearance of Angelica sourced from different origins, where discernible discrepancies in its visual characteristics among various production regions are limited. The challenge lies in the difficulty of manually extracting subtle distinguishing features, leading to reduced fitting capabilities of SVM, KNN, and decision tree algorithms. However, deep learning techniques excel in automatically extracting high-dimensional features through extensive data training, enabling effective identification of all eight production regions."
Comments on the Quality of English Language
Few places minor extensive corrections in grammar is required.
Response:
We have carefully revised the manuscript line by line. The revised words and sentences have been highlighted in red color in the revised manuscript.
Reviewer 3 Report
The authors used machine vision and deep learning to identification of Angelica sinensis. Some points are considered as follows:
1.What is the used method to dry A. sinensis samples?
2. What are the drying conditions for A. sinensis samples?
3.Please explain how dynamic data enhancement is often implemented using software tool functions in deep learning.
4. The authors wrote” CNN provides features extraction and image classification at the same time”. What are the extracted features which used to classify A. sinensis samples?
5. Please provide reference for Eqs (1)-(4).
6. Please give justifications, why do you select the initial learning rate training model to be 0.0001, 0.001, 0.01, 0.05 and 0.1?
7. Please give details about sdgm, adam and rmsprop algorithms.
8. Please give justifications, why do you select the kernel function to be Gaussian, and for the KNN, the number of nearest neighbors was set to 10.
9. The authors wrote” In this experiment, both offline and online dynamic data enhancement were used to amplify the data set and prevent over-fitting”, but in the results, these two data enhancement were not discussed.
10. What is the name of the used software to apply CNN and other algorithms, SVM, KNN, etc.?
Author Response
Dear Editor and reviewers:
We sincerely thank you and the anonymous reviewers for your helpful comments, corrections, and suggestions on the previous version of this manuscript. We believe that our work has benefited substantially from the invaluable input of the review team. Below is the detail of how we have addressed the reviewer’s comments.
All the changes within manuscript are highlighted in red colour, deleted words are not shown to avoid confusions. Moreover, the detailed corrections are listed below, point by point.
Refee#3
Comments and Suggestions for Authors
The authors used machine vision and deep learning to identification of Angelica sinensis. Some points are considered as follows:
1.What is the used method to dry A. sinensis samples?
Response:
Thank you very much for your suggestion. Our samples were collected from different regions and primarily dried, including natural sun drying, greenhouse drying, and hot air drying. Angelica sinensis from different regions not only exhibits variations in cultivation conditions but also undergoes different initial processing methods. Through long-term practice, it has been proven that Angelica sinensis from specific regions with suitable growing environments and excellent initial processing techniques (such as drying) achieves exceptional quality and is known as "Daodi" Angelica sinensis. Consequently, there are instances where other locations falsely claim to produce "Daodi" Angelica, necessitating the use of technical means to trace it back to its original source. This is precisely the significant purpose of our study. Therefore, the drying methods was not the focus of this study.
- What are the drying conditions for A. sinensis samples?
Response:
Thank you very much for your suggestion. We are only concerned about the origin of Angelica products and not the drying conditions of Angelica.
3.Please explain how dynamic data enhancement is often implemented using software tool functions in deep learning.
Response:
Thank you very much for your suggestion. We have added information about online data augmentation methods in lines 149-150: "Online enhancement is the random scaling (0.1-1.1 times) of the data during training to enhance the generalization of the model."
- The authors wrote” CNN provides features extraction and image classification at the same time”. What are the extracted features which used to classify A. sinensis samples?
Response:
Thank you very much for your suggestion. Thank you for your valuable suggestion. In deep learning, feature extraction takes place automatically during both training and testing stages. It can be conceptualized as a process occurring within a black box, with the ability to visualize the extracted features. However, the main emphasis of this paper is on identifying the origin of Angelica. Feature extraction in deep learning occurs automatically during both training and testing.
- Please provide reference for Eqs (1)-(4).
Response:
Thank you very much for your suggestion. We have made the necessary revisions accordingly, and the revision can be found in line 205.
- Please give justifications, why do you select the initial learning rate training model to be 0.0001, 0.001, 0.01, 0.05 and 0.1?
Response:
Thank you very much for your suggestion. We greatly appreciate your feedback. In our research, we have extensively consulted the data based on others researches and we added the related references in the revised manuscript. Specifically, we have relied on the findings from reference 37-38, which are cited in lines 238-239 of the article.
- Please give details about sdgm, adam and rmsprop algorithms.
Response:
Thank you for your suggestion. We have incorporated details regarding three optimization algorithms into lines 246-250 of the manuscript and referenced source 39-41. The specific additions are as follows:
"Sdgm is a stochastic gradient descent algorithm that utilizes momentum to reduce oscillation, expedite convergence, and escape local optima. Furthermore, rmsprop and Adam are adaptive learning rate optimizers that dynamically adjust the learning rate for each parameter. The adaptive adjustment enables gradual reduction of update steps for different parameters over time, leading to more precise parameter adjustments."
- Please give justifications, why do you select the kernel function to be Gaussian, and for the KNN, the number of nearest neighbors was set to 10.
Response:
Thank you for your valuable feedback. We have made corrections in lines 307-317 of the manuscript. The selection process involved extensive analysis and optimization using 10-fold cross-validation. Specifically, after comparing and evaluating the performance of four different kernel functions, namely linear, quadratic, cubic, and Gaussian, we determined that the Gaussian kernel function produced the best results. As a result, we chose it as the kernel function for SVM. Similarly, we conducted a thorough comparison of various numbers of nearest neighbors, including 1, 10, and 100. Our findings indicated that the optimal performance was achieved when the number of nearest neighbors was set to 10. Therefore, we selected 10 as the number of nearest neighbors for KNN.
- The authors wrote” In this experiment, both offline and online dynamic data enhancement were used to amplify the data set and prevent over-fitting”, but in the results, these two data enhancement were not discussed.
Response:
Thank you for your suggestion. In this study, we utilized both offline and online data augmentation methods in a sequential manner to expand the dataset, with the objective of mitigating the risk of overfitting caused by limited data volume. It is important to note that the offline and online data augmentation methods are not mutually exclusive alternatives, but rather complementary approaches that can be employed consecutively to enhance the overall effectiveness. Consequently, no direct comparison between these two methods was conducted in the study.
- What is the name of the used software to apply CNN and other algorithms, SVM, KNN, etc.?
Response:
Thank you for your valuable suggestions. All algorithms have been implemented using MATLAB 2021b software. We have incorporated the relevant content into lines 220-221.
Round 2
Reviewer 1 Report
Thank you for the corrections you have made. But there is one more small revision that you need to add to your study. It is important for readers and researchers who will reference your work to master the subject.
Please ask this question "When the number of samples given in Table 1 is examined, why is the number of samples of type SC small while the numbers of external samples of type SC are close to each other?" add the answer you have given to the study.
Author Response
Dear Editor and the reviewers:
We sincerely thank you and the anonymous reviewers for your helpful comments, corrections, and suggestions on the previous version of this manuscript. We believe that our work has benefited substantially from the invaluable input of the review team. Below is the detail of how we have addressed the reviewer’s comments.
All the new changes within manuscript are highlighted in red colour, deleted words are not shown to avoid confusions. Moreover, the detailed corrections are listed below, point by point.
Refee#1
Please ask this question "When the number of samples given in Table 1 is examined, why is the number of samples of type SC small while the numbers of external samples of type SC are close to each other?" add the answer you have given to the study.
Response:
Thanks for your good comments. In fact, there were apparent difference of the sample size among different origins. From beginning to end, the number of samples from SC was relatively low. We collected a total of 1,859 samples of Angelica sinensis and obtained 3,718 images (data1) containing both front and back images for each sample. During the data augmentation process, identical operations were conducted in the images obtained from all collection origins.
To make it clear, we had included a description about the image collection method in details in lines 143-144: "To capture all notable features, photographs of both the front and back sides of each sample were diligently taken." In addition, we had reorganized the text in lines 253-272 to make it clearer. Also, we had supplemented the image acquisition and preprocessing parts in lines 113-179.
Reviewer 2 Report
I appreciate the authors for having done some significant changes and addressing the comments given by the reviewers.
Author Response
Thanks so much. We had corrected it again.
Reviewer 3 Report
I accepted
Author Response

(The authors gave the same response as above.)
